# Health Belief, Behavior Intention, and Health Behaviors Related to Colorectal Cancer Screening in Taiwan

**DOI:** 10.3390/ijerph17124246

**Published:** 2020-06-14

**Authors:** I-Pei Lin, Ding-Tien Chung, Li-Yun Lee, Hsiang-Ju Hsu, Shu-Ching Chen

**Affiliations:** 1Department of Nursing, Ten-Chan General Hospital, Chung-Li, Taoyuan 320, Taiwan; r912055@gmail.com (I.-P.L.); hsiang336@yahoo.com.tw (H.-J.H.); 2Department of Family Medicine, Ten-Chan General Hospital, Chung-L, Taoyuan 320, Taiwan; denten@seed.net.tw; 3Department of Nursing, College of Nursing and Health Sciences, Da-Yeh University, Changhua 515, Taiwan; leeliyun@mail.dyu.edu.tw; 4School of Nursing and Geriatric and Long-Term Care Research Center, College of Nursing, Chang Gung University of Science and Technology, Taoyuan 333, Taiwan; 5School of Nursing, College of Medicine, Chang Gung University, Taoyuan 333, Taiwan; 6Department of Radiation Oncology and Proton and Radiation Therapy Center, Chang Gung Memorial Hospital at Linkou, Taoyuan 333, Taiwan

**Keywords:** colorectal cancer screening, health belief, behavior intention, health behavior

## Abstract

Health belief and behavior intention affect subsequent health behaviors. The purpose of this study was to assess the levels of health belief, behavior intention, and health behavior, and to identify the factors related to health behaviors in adults receiving colorectal cancer (CRC) screening in Taiwan. This cross-sectional study recruited patients receiving a CRC screening from the cancer screening outpatient department of a teaching hospital in northern Taiwan. Demographic and health characteristics were recorded, and participants were assessed using Champion’s health belief model scale, cancer screening intention scale, and the health protective behavior scale. Of the 125 subjects (aged 49–75 years), 27.2% reported active screening; the rest passively received screening after doctor referral. Those who were doctor-referred had lower levels of health behavior, including general behavior, self-knowledge, and health care. Positive health behaviors related to CRC screening were associated with not smoking, greater seriousness in health belief, more confidence in health belief, consuming the recommended amount of fruits and vegetables, and motivation for CRC screening; these factors explained 35.0% of the variance in positive health behaviors related to CRC screening. A comprehensive education program encouraging CRC cancer screening should include access to available resources and encourage positive health belief and behavior intention related to this important cancer screening activity.

## 1. Introduction

Globally, colorectal cancer (CRC) is the third most commonly diagnosed malignancy and the second leading cause of cancer death [1]. In Taiwan, CRC is the second most common cancer and the third leading cause of cancer mortality [2]. Western lifestyles increase the risk of CRC [3]. The rate of CRC has increased in recent years, especially rapidly in adults older than 50 years [1,2]. Screening for CRC using the fecal occult blood test (FOBT) can detect precursor lesions and reduce the incidence of malignant CRC [4]. Adults aged 50 years or more are recommended to receive a FOBT annually [5]. In Taiwan, four types of cancer screening (FOBT, oral mucosal screening, mammography, and pap smear screening) have been a part of public health policy since 2004 and free cancer screening is provided for those who meet screening criteria [6]. The FOBT screening rate of around 40% is lower than that of Western countries [7].

Negative health beliefs and less behavior intention to obtain CRC screening result in a reduced likelihood of receiving FOBT. Belief is an idea that a person holds to be true which can modify behavior through experiences, culture, societal norms, or education [8]. An individual’s beliefs about a health problem, perceived benefits, barriers, and self-efficacy all determine whether or not that person will engage in health promoting behaviors [8]. Behavior intention refers to a person’s subjective determination and the probability that the individual will perform some behavior [9]. Those who have a positive attitude and belief toward cancer screening [10], perceive the issue as more serious [11], perceive more benefits to screening [12], have a lower barrier to screening [10,12], and receive advice to screen from healthcare providers [12] are all more likely to have the intention to screen.

Health behavior is any activity undertaken by a person who believes himself/herself to be healthy for the purpose of preventing disease or detecting disease at an asymptomatic stage [13]. People with lower household income [14], those who lack health insurance [14], smokers [14], those with no family history of the specific cancer [15], those with a negative belief related to the screening [16], those with more barriers to screening [16], those with more fear [15], as well as those with less susceptibility, seriousness, health motivation, and perception of self-efficacy [15], have lower adherence to cancer screening. People with unfavorable beliefs and less intention to screen have health behaviors that prevent them from receiving CRC screening. Based on a review of the literature, we assumed that people who had lower level of education, less healthy lifestyles (e.g., smoking, drinking, and insufficient intake of fruits and vegetables), lack of screening motivation (e.g., doctor referral rather than self-referral), and negative health belief would be less likely to exhibit positive health behaviors. Therefore, the purpose of this study was to (1) assess the levels of health belief, behavior intention, and health behavior and (2) identify the factors associated with health behaviors among people receiving CRC screening.

## 2. Materials and Methods

### 2.1. Design and Sample

A descriptive cross-sectional study was conducted with a convenience sample. Data were collected from April to September 2019. Participants were recruited from the cancer screening outpatient department of a teaching hospital in northern Taiwan. The inclusion criteria were: (1) age ≥ 50 years (the age for CRC screening according to the Health Promotion Administration, Ministry of Health and Welfare, Taiwan policy), (2) received FOBT, (3) ability to speak/read in Mandarin and Taiwanese, and (4) agreement to participate in the study after explanation of its purposes and procedures.

### 2.2. Ethical Considerations

The ethical review board of the study institution approved the study (Number: 108-B-03-01), which was conducted in accordance with the Declaration of Helsinki, and written informed consent was obtained from all participants after detailed explanation of the study goals and procedures.

### 2.3. Data Collection

Participants with a doctor referral or who met CRC screening criteria and actively asked for CRC screening or regularly visited the cancer screening outpatient department were recruited. A well-trained research nurse helped the participants complete the questionnaires, which took around 10–15 min.

### 2.4. Champion’s Health Belief Model Scale (CHBMS)

Champion’s health belief model scale (CHBMS) was used to assess the health beliefs related to CRC screening [17]. This 36-item scale has six subscales and an additional subscale asking participants to comment on the instrument itself: susceptibility (5 items), seriousness (7 items), benefits (6 items), barriers (6 items), health motivation (4 items), confidence (5 items), and scale items (3 items). Items are scored on a Likert scale ranging from 1 (strongly disagree) to 5 (strongly agree), with higher scores indicating a greater positive belief about CRC screening. Previous studies have demonstrated satisfactory psychometric characteristics for this instrument [17,18]. In the present study, the Cronbach’s α was 0.92.

### 2.5. Cancer Screening Intention Scale (CSIS)

Participants’ intention to screen for CRC were assessed using the cancer screening intention scale (CSIS) [19]. The instrument consists of 18 items, with responses scored on a Likert scale from 1 (strongly disagree) to 4 (strongly agree), with higher scores indicating greater intention to receive CRC screening. The CSIS has been demonstrated to be reliable in terms of CRC screening [19]. In the present study, the Cronbach’s α was 0.82.

### 2.6. Health Protective Behavior Scale (HPBS)

Individual’ health protective behaviors and health promoting behavior were assessed using the health protective behavior scale (HPBS) [20]. This 32-item scale has five subscales: interpersonal support (8 items), general behavior (7 items), self-knowledge (6 items), nutrition behavior (5 items), and health care (6 items). Responses are scored on a 1 to 5 scale, and higher scores indicate more positive health behaviors. Satisfactory psychometric properties were reported in previous studies [17,20]. Cronbach’s αs for the five subscales ranged from 0.72 to 0.96 in the present study.

### 2.7. Demographic and Health Characteristics

A sheet of demographic and health characteristics was used to collect information including age, gender, education level, body mass index, religion, annual family income (in New Taiwan dollars, NT), status of chewing betel nut, smoking, drinking, family history of CRC, motivation for CRC screening, prior CRC screening, results of CRC screening, and amount of fruit and vegetable intake (recommended amount or not).

### 2.8. Data Analysis

Data were analyzed with SPSS, version 21.0 for Windows (IBM Corp., Armonk, NY, USA). Demographic characteristics, health characteristics, health belief, behavior intention, and health behavior were analyzed using descriptive statistics (frequency distribution, percentage, means, and standard deviations [SDs]). Independent-samples *t*-test was used to compare health belief, intention, and health behavior between the doctor referral group and the active screening group. Stepwise linear regression analysis was conducted to identify factors related to health behavior. The independent variables selected included education level (years), prior CRC screening (no vs. yes), smoking (no vs. yes), drinking (no vs. yes), meeting the recommended intake of fruits and vegetables (no vs. yes), motivation for CRC screening (doctor referral vs. active screening), health belief, and intention.

## 3. Results

### 3.1. Subject Demographic and Health Characteristics

Of the 125 eligible subjects approached, all completed the questionnaire with a response rate of 100%. The mean age of patients was 62.38 (SD = 6.9) years, ranging 50–75 years. More than half of the subjects were male (n = 64, 51.2%). In terms of socioeconomic status, 60.0% (n = 75) of subjects were unemployed and 9.6% (n = 12) were skilled workers. Most participants were married (n = 112, 89.6%); educated at the junior-high (n = 27, 21.6%) or senior-high (n = 46, 36.8%) level; reported their religion as Buddhism or Taoism (n = 80, 64.0%); and had an average family annual income less than NT 499,999 (n = 86, 68.8%). Most participants reported no family history of CRC (n = 109, 87.2%), were non-smokers (n = 103, 82.4%), were non-drinkers (n = 114, 91.2%), had normal results (n = 118, 94.4%) or hemorrhoids combine polyps (n = 5, 4.0%) identified at CRC screening, had a CRC screening because of doctor referral (n = 91, 72.8%), had no prior CRC screening (n = 73, 58.4%), had met the recommended intake of fruits and vegetables (n = 110, 88%), and had a body mass index greater than 18.5 kg/m^2^ (Table 1).

### 3.2. Levels of Health Belief, Intention, and Health Behavior

The mean score for health belief was 116.82 (SD = 19.33). The mean scores for the subscales were as follows: for susceptibility, 10.60 (SD = 5.43); for seriousness, 22.06 (SD = 8.34); for benefits, 24.86 (SD = 4.72); for barrier, 12.90 (SD = 4.68); for confidence, 20.01 (SD = 3.35); and for health motivation, 15.50 (SD = 5.88). The mean score for intention to receive CRC screening for the participants was 56.29 (SD = 5.88). The mean score for health behavior was 110.30 (SD = 16.35). The mean scores for the subscales of health behavior were as follows: for interpersonal support, 25.98 (SD = 5.91); for general behavior, 22.58 (SD = 5.02); for self-knowledge, 22.89 (SD = 3.32); for nutrition behavior, 17.78 (SD = 4.03); and for health care, 21.08 (SD = 4.49) (Table 2).

### 3.3. Health Belief, Intention, and Health Behavior Comparison between the Doctor Referral Group and the Active Screening Group

Of the 125 participants, 91 received a CRC screening because of doctor referral and 34 received a CRC screening because of active screening. No statistically significant differences were noted in health belief and intention between those who were doctor-referred and those who received active screening. Those who were doctor-referred had lower levels of overall health behavior, general behavior, self-knowledge, and health care. These differences were statistically significant (Table 3).

### 3.4. Factors Associated with Health Behavior

We performed stepwise linear regression analysis to identify the factors significantly associated with health behavior (Table 4). Participants who were non-smokers (β = −0.273), had a higher level of seriousness in their health belief (β = 0.347), had a higher level of confidence in their health belief (β = 0.256), had the recommended intake of fruits and vegetables (β = 0.232), and had a CRC screening because of active screening (β = 0.180) were more likely to have CRC-screening health behavior. These five factors explained 35.0% of the total variance in this parameter.

## 4. Discussion

The study examined health belief, behavior intention, and health behavior in relation to CRC screening. Our subjects’ health belief was similar to that reported in previous studies [21,22]. However, compared to one study of belief involving breast cancer screening in northwest Iran, we found a higher level of benefit belief and a lower level of barrier belief than that found for women undergoing mammography [22]. This difference may be due to a difference in the time available to do cancer screening. Most of the subjects recruited in the study in northwest Iran were housewives, but more than one third of our subjects were employed. Perhaps the CRC screening process is more complicated than that for breast cancer screening. When healthcare providers assess the CRC screening belief of each individual, provide education, and discuss in detail the procedure of the FOBT for eligible people, subjects can better understand the need for early detection of precursor lesions.

The current study showed that 73.6% of participants passively received CRC screening after visiting the outpatient clinic and being referred by their attending physicians after meeting the CRC screening criteria. Prior studies reported that people with low socioeconomic status, including low income, education, or low insurance coverage, are less likely to undergo cancer screening [23,24]. In Taiwan, CRC screening is one of four major cancer screenings freely available under the National Health Insurance (NHI) for virtually all residents [6]. This result likely reflects the widespread concerns with obtaining CRC screening. Support should be provided to interventions that address screening barriers such as inconvenience, limited information (FOBT procedure), and knowledge deficit (asymptomatic no need to screen). Health professionals should be educated to take an active role in offering screening guidelines during medical encounters with eligible patients.

The present study demonstrated that people who smoked were more likely to have negative health behaviors in relation to CRC screening. This finding is consistent with those of a prior study which indicated that women who did not drink alcohol, did not smoke, and had physical activity tended to also show higher adherence to breast cancer screening [25]. Smoking increases CRC risk [26]. This result indicates that cancer prevention, screening, and early detection programs for smokers may help them develop more positive health behaviors.

One major finding was those who were taking in the recommended amount of fruits and vegetables were more likely to also have positive health behavior. However, most of the subjects in this current study met the recommended guidelines for fruit and vegetable intake. In Taiwanese adults, 20% had the recommended number of servings for fruit and 30% met the Taiwan Food-Guide recommendations for vegetables [27]. Studies show that fruits and vegetables contain a large number of vitamins, minerals, and dietary fiber, and insufficient intake of fruits and vegetables is associated with a higher risk of CRC [28,29]. The present study suggests the need for a targeted approach to help adults with insufficient fruit and vegetable intake to increase their positive health behaviors in terms of diet and adhere to CRC screening.

Our results indicated that those who actively asked for CRC screening reported higher levels of health belief, intention, and health behaviors than those who were doctor-referred. This difference can probably be explained by the fact that people had have a positive attitude and belief toward cancer screening [10] and have more intention to screen are more likely to take some precautions. However, the sample recruited people receiving CRC screening, and those who do not receive CRC screening may differ in their health belief, intention, and health behaviors. Motivation for CRC screening may influence the health belief, intention, and health behaviors of subjects, subsequently affecting the results of the study. Future studies may expand subject recruitment to different populations in order to compare the effect of screening test variation on belief and intention to screen.

## 5. Conclusions

In this study, 26.4% of participants reported active screening. Those who were doctor-referred had lower levels of health behavior, including general behavior, self-knowledge, and health care. People who did not smoke, perceived more seriousness in their health belief, had more confidence in their health belief, ate the recommended amount of fruits and vegetables, and actively requested CRC screening were more likely to have positive health behaviors. The results of this study provide a reference for clinical assessment of health belief and behavior intention and the factors associated with health behavior in people receiving CRC screening.

### 5.1. Limitations and Future Research

This study had several limitations. First, our study recruited patients who received a FOBT for CRC screening and excluded those did not have a CRC screening. The health beliefs, behavior intention, and health behaviors may differ between these two populations. The ability to generalize results to those without a CRC screen is limited. Motivation for CRC screening may differ by health belief and reason for screening, so comparative studies are needed to more completely identify health behavior issues in these patients. Second, we did not consider immigrant status or interruption in NHI coverage as factors in refusing cancer screening. Thus, future studies should expand recruiting to those with different insurance status and explore the barriers to undergoing cancer screening. Finally, this study recruited people over 50 years of age, and young adults those who were less than 50 years not included National Health Insurance pays for CRC screening. Although the national policies of Taiwan concerning cancer screening include CRC screening, future studies are still needed to develop or modify the health policies elsewhere in the country.

### 5.2. Clinical Implications

A comprehensive education program of CRC screening should incorporate adults with health inequality. Providing accessible resources such as mobile CRC screening vehicles or screening by mail are recommended to help encourage people to adopt positive health behaviors in terms of preventive measures such as CRC screening.

## Figures and Tables

**Table 1 ijerph-17-04246-t001:** Demographic and health characteristics of subjects (N = 125).

Variable	Number (%)	Mean (SD)	Range
Age		62.38 (6.9)	50–75
Gender			
Male	64 (51.2)		
Female	61 (48.8)		
Social economic status			
Unemployed	75 (60.0)		
Unskilled/semi-skilled worker	6 (4.8)		
Skilled worker	12 (9.6)		
Clerk, shop, owner, farm owner	2 (1.6)		
Semi-profession	5 (4.0)		
Profession	2 (1.6)		
Other	23 (18.4)		
Marital status			
Unmarried	13 (10.4)		
Married	112 (89.6)		
Educational level			
None	3 (2.4)		
Elementary	23 (18.4)		
Junior high	27 (21.6)		
Senior high	46 (36.8)		
College and above	26 (20.8)		
Religion			
None	41 (32.8)		
Buddhism/Taoism	80 (64.0)		
Christianity/Catholicism	3 (2.4)		
Other	1 (0.8)		
Family annual income (NT)			
<499,999	86 (68.8)		
500,000–1,000,000	34 (27.2)		
>1,000,000	5 (4.0)		
Family history of colorectal cancer			
Nil	109 (87.2)		
Yes	16 (12.8)		
Smoking			
Nil/ex-smoker	103 (82.4)		
Current smoker	22 (17.6)		
Drinking			
Nil/ex-smoker	114 (91.2)		
Current smoker	11 (8.8)		
Results of CRC screening			
Normal	118 (94.4)		
Hemorrhoids	1 (0.8)		
Hemorrhoids combine polyps	2 (1.6)		
Positive untracked diagnosis	1 (0.8)		
Polyps	1 (0.8)		
Colitis	0		
Colorectal adenomas	0		
Motivation for CRC screening			
Active screening	34 (27.2)		
Doctor referral	91 (72.8)		
Prior CRC screening			
No	52 (41.6)		
Yes	73 (58.4)		
Recommended intake of fruits and vegetables			
No	15 (12.0)		
Yes	110 (88.0)		
Body Mass Index			
≥18.5 kg/m^2^	123 (98.4)		
<18.5 kg/m^2^	2 (1.6)		

Abbreviations: SD, standard deviation; NT, New Taiwan Dollars; CRC, colorectal cancer.

**Table 2 ijerph-17-04246-t002:** Scores for health belief, intention, and health behavior (N = 125).

Variable	Mean	SD	Range	Theoretical Scoring Range
Health belief (CHBMS)	116.82	19.33	70–180	0–180
Susceptibility	10.60	5.43	5–25	0–25
Seriousness	22.06	8.34	7–35	0–35
Benefits	24.86	4.72	6–30	0–30
Barrier	12.90	4.68	6–30	0–30
Confidence	20.01	3.35	7–25	0–25
Health motivation	15.50	2.93	6–20	0–20
Intention (CSIS)	56.29	5.88	30–70	18–72
Health behavior (HPBS)	110.30	16.35	58–110	32–160
Interpersonal support	25.98	5.91	8–40	0–40
General behavior	22.58	5.02	7–35	0–35
Self-knowledge	22.89	3.32	17–30	0–30
Nutrition behavior	17.78	4.03	7–25	0–25
Health care	21.08	4.49	12–30	0–30

Abbreviations: SD, standard deviation; CHBMS, Champion’s health belief model scale, theoretical scoring range 0–180; CSIS, cancer screening intention scale, theoretical scoring range 18–72; HPBS, health protective behavior scale, theoretical scoring range 18–72.

**Table 3 ijerph-17-04246-t003:** Health belief, intention, and health behavior of the doctor referral group and the active screening group (N = 125).

Characteristics	Doctor Referral Group (N = 91)	Active Screening Group (N = 34)	*t*	*p*
	Mean (SD)	Mean (SD)		
Health belief (CHBMS)	115.40 (20.00)	120.62 (17.08)	−0.686	0.494
Susceptibility	5.33 (0.56)	5.76 (0.99)	−0.236	0.814
Seriousness	21.96 (8.41)	22.35 (8.25)	−1.657	0.100
Benefits	24.44 (5.02)	26.00 (3.64)	1.009	0.315
Barrier	13.15 (4.70)	12.21 (4.62)	−1.861	0.065
Confidence	19.67 (3.48)	20.91 (2.82)	−1.226	0.222
Health motivation	15.31 (3.04)	16.03 (2.59)	−1.349	0.180
Intention (CSIS)	55.92 (6.15)	57.26 (5.05)	−1.136	0.258
Health behavior (HPBS)	107.65 (14.91)	117.41 (18.07)	−3.071	0.003
Interpersonal support	25.49 (5.87)	27.26 (5.91)	−1.498	0.137
General behavior	21.80 (4.85)	24.65 (4.96)	−2.903	0.004
Self-knowledge	22.36 (3.20)	24.29 (3.28)	−2.983	0.003
Nutrition behavior	17.44 (4.01)	18.71 (4.00)	−1.573	0.118
Health care	20.55 (4.17)	22.50 (4.74)	−2.240	0.027

Abbreviations: SD, standard deviation; CHBMS, Champion’s health belief model scale; CSIS, cancer screening intention scale; HPBS, health protective behavior scale.

**Table 4 ijerph-17-04246-t004:** Factors significantly associated with health behavior based on multiple regression analysis (N = 125).

Variable	ß	*p*	95% CI		Adjusted R^2^	F
			Lower	Upper		
					0.350	12.817
Smoking (no vs. yes)	−0.273	0.001	−18.061	−5.287		
Health belief—seriousness (CHBMS)	0.347	0.001	0.983	0.376		
Health belief—confidence (CHBMS)	0.256	0.002	0.484	2.015		
Consuming the recommended amount of fruits and vegetables (no vs. yes)	0.232	0.003	4.168	19.118		
Motivation for CRC screening (doctor referral vs. active screening)	0.180	2.374	1.089	12.047		
Constant	------	0.001	73.875	106.768		

Abbreviations: CI, confidence interval; CHBMS, Champion’s health belief model scale. Input independent variable: covariates included education level (year) (continuous score), prior CRC screening (no vs. yes), smoking (no vs. yes), drinking (no vs. yes), consuming the recommended amount of fruits and vegetables (no vs. yes), motivation for CRC screening (doctor referral vs. active screening), health belief (continuous score), and intention (continuous score).

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
