# Peer review of "Health Belief, Behavior Intention, and Health Behaviors Related to Colorectal Cancer Screening in Taiwan"

_ijerph, 2020, doi:10.3390/ijerph17124246_

Round 1

Reviewer 1 Report

Dear Author/s,

Thank you for your revision of the article.

I would recommend to bring "Limitation and future research" sub-section to bring at the end of the conclusion.

"Conclusion" require major revision and need to explain it more adequately by comparing/ contrasting with existing literature as well as by HIGHLIGHTING the contributions that this paper might make to the current literature.

Author Response

Point 1: I would recommend to bring "Limitation and future research" sub-section to bring at the end of the conclusion.

Response 1: Thank you for your comments. We have moved this paragraph at the end of conclusion (page 8, lines 216–227, and revised text marked in red words).

Point 2: "Conclusion" require major revision and need to explain it more adequately by comparing/ contrasting with existing literature as well as by HIGHLIGHTING the contributions that this paper might make to the current literature.

Response 2: Thank you for your suggestions. We have revised and added the contributions of this current study (page 8, lines 212–215, and revised text marked in red words).

Reviewer 2 Report

Thank you for the opportunity to review the manuscript

The topic of the paper is of relevance for policy makers, and could develop into an interesting paper with more work. Unfortunately, in my opinion, in the current form the paper does not meet the journal's threshold for publication.

Following are some reactions I had from reading this manuscript, that I hope can contribute to improve it.

The authors state three main goals; (1) assess the level of health belief, behaviour intention, and health behaviour, and (2) Identify the factors associated with health behaviours among people receiving CRC screening. For that they use a convenience sample that includes individuals with a doctor referral or those who have met the CRC screening criteria and actively asked for. 

They use three standardised surveys: “Champion Health Belief Model Scale” (CHBM), Cancer Screening Intention Scale (CSIS)” “Health Protective Behavior Scale” ( HPBS).

If understood, the variable of interest is Health behaviour (HPBS). The variable was elicited among a sample of individuals who have performed a CCR screening, but since the counterfactual is not individuals who did not ( because refused to do it or because doctors select patients), it is difficult to relate it with the CCR screening decision. So, in my opinion, the authors cannot state “The study examined health belief, behaviour intention, and health behaviour in relation to CRC screening”, or identify the variable as “CRC screening health behaviour?”. 

In the same vein, the authors do not justify why the sample was restricted to individuals receiving CRC screening. A more relevant policy questions would be understanding health behavioural differences between individuals who adhere or not to CCR screening. The authors could also explore the differences between individuals that follow the doctor referral and those who actively asked for CRC screening.

The authors also need to clarify the methods. Which multiple regression were applied? One should note that OLS regression should not be applied to a bounded variable.  

The authors state they estimated Pearson coefficient but they do not properly report it. Moreover, the authors should be cautious while interpreting the statistical evidence since they are using a convenience sample.

It is not clear to me how to interpret the intentions to CRC screening among a population that have already done it.

I would like to stress that the paper. Goals are interesting and that the paper would benefit from the authors exploring more all the data they have collected and be more careful with their interpretation.  Understand  health beliefs, behaviour intentions and health behaviour is relevant for policy. They have to make clear that data was collected among this group of individuals but that there is no evidence that can be associated with their  screening decision. 

The paper would benefit from a discussion how individuals can access to the screening in particular without be referred by a doctor.

Author Response

Point 1: In the same vein, the authors do not justify why the sample was restricted to individuals receiving CRC screening. A more relevant policy questions would be understanding health behavioural differences between individuals who adhere or not to CCR screening. The authors could also explore the differences between individuals that follow the doctor referral and those who actively asked for CRC screening.

Response 1: Thank you for your comments. We have added the differences in health belief, intention, and health behavioural between the doctor referral and those who actively asked for CRC screening (page 6–7, lines 155–163, and revised text marked in red words).

Point 2: The authors also need to clarify the methods. Which multiple regression were applied? One should note that OLS regression should not be applied to a bounded variable.

Response 2: Thank you for your suggestions. We have revised the methods of data analysis (page 3, lines 110–113, and revised text marked in red words).

Point 3: The authors state they estimated Pearson coefficient but they do not properly report it. Moreover, the authors should be cautious while interpreting the statistical evidence since they are using a convenience sample.

Response 3: Thank you for your comments. We have added a paragraph of hypothesis (page 2, lines 57–59, and revised text marked in red words) and revised the methods of data analysis (page 3, lines 110–113, and revised text marked in red words).

Point 4: It is not clear to me how to interpret the intentions to CRC screening among a population that have already done it.

Response 4: Thank you for your suggestions. We have added the differences in health belief, intention, and health behavioural between the doctor referral and those who actively asked for CRC screening (page 6–7, lines 155–163, and revised text marked in red words).

We also have added a paragraph of discussion for this issue (page 8, lines 200–208, and revised text marked in red words).

Point 5: I would like to stress that the paper. Goals are interesting and that the paper would benefit from the authors exploring more all the data they have collected and be more careful with their interpretation.  Understand  health beliefs, behaviour intentions and health behaviour is relevant for policy. They have to make clear that data was collected among this group of individuals but that there is no evidence that can be associated with their screening decision. 

Response 5: Thank you for your suggests. We have listed this issue as a limitation in this current study (page 8, lines 220–221, and revised text marked in red words).

Point 6: The paper would benefit from a discussion how individuals can access to the screening in particular without be referred by a doctor.

Response 6: Thank you for your comments. We have added a paragraph of discussion for this issue (page 8, lines 200–208, and revised text marked in red words).

Round 2

Reviewer 2 Report

Thank you for the opportunity of reviewing the revised paper.  I believe tha authors have substantially  improved the paper

Two minor suggestions:

The section Health Belief, Intention, and Health Behavior between Doctor Referral Group and Active Screening Group, should come before multiple regression analysis

The authors should make clear they are estimating an stepwise linear regression. 

Author Response

Point 1: The section Health Belief, Intention, and Health Behavior between Doctor Referral Group and Active Screening Group, should come before multiple regression analysis.

Response 2: Thank you for your suggestions. We have move the section health belief, intention, and health behavior between doctor referral group and active screening group before multiple regression analysis (page 6–7, lines 142–165, and revised text marked in red words). We also changed the both paragraphs in the abstract, data analysis, and conclusion section (page 1, lines 21–23, page 3, lines 109–110,  page 8, lines 212–213, and revised text marked in red words).

Point 2: The authors should make clear they are estimating an stepwise linear regress.

Response 1: Thank you for your comments. We have revised the method of data analysis as “stepwise linear regression” following your comment (page 3, line 109, page 6, line 142, and revised text marked in red words).